

# Prevalence of intestinal parasites and comparison of detection techniques for soil-transmitted helminths among newly arrived expatriate labors in Jeddah, Saudi Arabia

Mohammad F. Al-Refai[1] and Majed H. Wakid[2,3]

[1] Medical Microbiology, Department of Medical Laboratories, King Fahd Armed Forces Hospital, Jeddah, Saudi Arabia
[2] Faculty of Applied Medical Sciences, Medical Laboratory Sciences Department, King Abdulaziz University, Jeddah, Saudi Arabia
[3] Special Infectious Agents Unit, King Fahd Medical Research Center, Jeddah, Saudi Arabia

Corresponding author
Majed H. Wakid,
mwakid@kau.edu.sa

## ABSTRACT

**Background:** Diversity in clinical signs and symptoms are associated with soil transmitted diseases (STD), which are spread to humans by intestinal worms and transmitted in a variety of ways. There is a need for the present study, which aimed to investigate the prevalence of intestinal parasites and to compare between the common detection techniques for soil-transmitted helminths (STHs) among newly arrived expatriate labors in Jeddah, Saudi Arabia.

**Methods:** A total of 188 stool samples were analyzed by macroscopic examination, and microscopic examination using direct iodine smear and the formal ether sedimentation technique. Trichrome and modified Kinyoun's stains were used to confirm the morphology of any detected protozoa stages and oocyst of *Cryptosporidium*, respectively. A chromatographic immunoassay kit was used for *Entamoeba histolytica*, *Giardia lamblia* and *Cryptosporidium*. In addition, real-time PCR was employed only to identify various STHs.

**Results:** Out of 188, several types of parasites were detected in 35 samples (18.62%), of which some with multiple infections. Nine samples (4.79%) were positive for *Entamoeba coli*, seven samples (3.72%) for *Trichuris trichiura*, six samples (3.19%) for *Necator americanus*, four samples (2.13%) for *Strongyloides stercoralis*, four samples (2.13%) for *Ascaris lumbricoides*, four samples (2.13%) for *E. histolytica*, three samples (1.60%) for *Blastocystis hominis* and two samples (1.06%) for *Ancylostoma duodenale*. In comparison between laboratory techniques for STHs, real-time PCR was able to detect the DNA of 19 samples (10.1%) followed by Ritchie sedimentation technique (18, 9.6%), and direct smear (7, 3.7%) ($p > 0.05$).

**Conclusion:** The high rate of newly arrived foreign workers infected with intestinal parasites could lead to a risk to society. Continuous and regular surveys are needed to deal with the occurrence of intestinal parasitic infections including STHs.
To improve the identification of these infections, we recommend a supporting infrastructure for the application of concentration methods and molecular assays.

# INTRODUCTION

Infectious intestinal parasites, which include helminths and protozoans, may cause several serious health problems such as malnutrition, anemia, and cancer, especially in impoverished and tropical countries (*World Health Organization, 2002*; *Torgerson et al., 2015*). More serious complications are seen in immunocompromised patients (*Bahwaireth & Wakid, 2022*; *Omrani et al., 2015*). Several serious waterborne outbreaks in the world were caused by parasites (*Efstratiou, Ongerth & Karanis, 2017*).

According to the *World Health Organization (2020)* (WHO), in 2020 roughly 25% of the global population was infected with soil-transmitted helminths (STHs), while over 3 billion revealed no symptoms. These STHs belong to the Phylum Nematoda (roundworms), and include *Ascaris lumbricoides*, hookworm (*Ancylostoma duodenale* and *Necator americanus*), *Strongyloides stercoralis* and *Trichuris trichiura*. Contaminated soil plays a vital role in STHs transmission and life cycles (*Wakid, 2020b*).

*Blastocystis hominis, Entamoeba histolytica Giardia lamblia*, and *Cryptosporidium* are the common intestinal protozoan parasites. According to the WHO, in the developing world, *G. lamblia* is the most pervasive parasite that causes diarrhea, while the invasive amoebic infection affects approximately 50 million people every year, resulting in 40–100 thousand deaths (*World Health Organization, 1997*). The majority of intestinal protozoa spread through fecal contamination caused by poor sanitation and contaminated water or food (*World Health Organization, 2006*; *CDC, 2019*; *Cama & Mathison, 2015*).

Infection with non-pathogenic intestinal protozoa such as *E. coli, Endolimax nana* and *Iodamoeba buetschlii* should be reported, which indicates a lack of hygiene, as their route of transmission is similar to the pathogenic protozoa.

The increased risk of getting intestinal parasites is due to a variety of socioeconomic, environmental, and hygienic variables. Rapid socioeconomic progress and long-term economic stability, on the other hand, have resulted in a massive migration of expatriate labor, primarily from less wealthy and developed nations (*Baker et al., 2022*). Within a population, parasitic infection patterns vary and are connected to origin nations (*Blair & Sharif, 2012*; *Arfaa, 1981*; *Salas, Heifetz & Barrett-Connor, 1990*; *Varkey et al., 2007*; *Abu-Madi, Behnke & Ismail, 2008*), host gender (*Stephenson, Latham & Ottesen, 2000*), and yearly changes in parasite transmission rates within a population pool (*Varkey et al., 2007*; *Wang, 1998*; *Al-Shammari et al., 2001*).

The Kingdom of Saudi Arabia (KSA) is a fast-expanding country with a diverse population from a wide range of educational backgrounds, religious views, eating and leisure habits and behaviors, and cultural customs. In 2021, the KSA's population was around 34 million, with a growth rate of 1.5% (*General Authority for Statistics, 2021*; *The World Bank, 2021*). The majority of expatriates to the KSA come from underdeveloped nations in Africa and South Asia (*UNICEF, 2014*; *United Nation, 2021*), where parasitic

illnesses are common (*Cross & Basaca-Sevilla, 1981*; *Ward, 2009*; *Abubakar, Tillmann & Banerjee, 2015*). Over 80% of the workforce in the KSA is made up of expatriates from Asian countries (*Expatriate Population, 2018*).

The diagnosis of intestinal parasites including STHs is mainly based on the microscopic examination of stool samples by using different techniques. Recently, the molecular diagnosis by real-time PCR has provided a more sensitive and specific laboratory method (*World Health Organization, 2019*; *Garcia et al., 2017*).

Using improved and cost-effective diagnostic tools as a routine diagnostic strategy helps to minimize the under-diagnosis and underreporting of intestinal parasites including STHs infections. This will be supportive for more feasible preventive and control measures. Therefore, the aim of this study was to investigate the prevalence of intestinal parasites, and to compare the microscopic (direct smears, sedimentation concentration) and molecular (real-time PCR) techniques for the detection of STHs among newly arrived expatriate labors in Jeddah, Saudi Arabia.

## MATERIALS AND METHODS

This study was conducted according to the guidelines of the Declaration of Helsinki and approved by The Ethics and Research Committee of the Faculty of Applied Medical Sciences, King Abdulaziz University, (FAMS-EC2018-012), and written informed consent was obtained from each participant.

### Samples collection and study group

This cross-sectional study included newly arrived expatriate labors in Jeddah regardless of gender and age. Stool specimens were collected from a laboratory of expatriate workers polyclinic between January and March 2018. Each participant was handled a container and guidelines to collect the stool specimen. A total of 210 specimens were collected, but after checking the quantity of the stool, 22 specimens were excluded from the study due to insufficient amounts. According to that, a total of 188 specimens were used for this study.

The collected samples were refrigerated, and then investigated through gross examination, light microscopy and real-time PCR on a daily basis.

The laboratory investigation was conducted at the Diagnostic Parasitology Laboratory in the Special Infectious Agents, King Fahd Medical Research Centre at King Abdulaziz University in Jeddah, Saudi Arabia.

### Macroscopic examination

Stool specimens were examined grossly to determine the consistency (hard, formed, loose or watery), color and presence of blood, mucus, and gross parasitic stages such as segments of tapeworms or whole adult worms.

### Microscopic examination by direct smears

For a direct smear, about 1–2 mg of stool was mixed by wooden stick with 1–2 drops of Lugol's' iodine on a clean glass slide, then mixed and covered by a 22 mm × 22 mm coverslip. The slide was examined under 10× and 40× objective lenses of the light microscope.

### Microscopic examination by sedimentation technique

Ritchie formal-ether sedimentation technique was performed by emulsifying 2 gm of stool in 10 ml of 10% formal saline. The suspension filtered *via* two layers of gauze fitted to a funnel into a 15 ml centrifuge tube, then centrifuged for 5 min at 2,000 rpm. The supernatant was discarded, and the sediment was re-suspended with 10 ml of formal saline (10%). After that, 3 ml of diethyl ether was added, and the tube was shaken vigorously for 15 s, and then centrifuged for 5 min at 2,000 rpm. Four layers were observed, the diagnostic stages were precipitated at the bottom of the tube, and the debris formed a layer separated between diethyl ether and formal saline. A wooden stick was used to detach the debris layer, and the tube rapidly inverted to decant the unwanted layers. Finally, the sediment was mixed with two drops of iodine and a drop was placed on a microscope slide and covered with 22 mm × 22 mm cover glass, then examined under 10× and 40× objective lenses.

### Microscopic examination by trichrome permanent staining

To confirm the morphology of the protozoan parasites, PVA-preserved stool samples were stained with Para-Pak® trichrome kit, according to the manufacturer's instructions, and then examined microscopically by oil immersion objective lens as described previously (*Aldahhasi, Toulah & Wakid, 2020*).

### Microscopic examination by modified kinyoun's staining

This stain was used to detect *Cryptosporidium*. A stool smear was prepared from each sample and allowed to air dry, then fixed with methanol. After that, the smear was stained for 5 min with carbol-fuchsin. Each smear was then washed with tap water, decolorized for 2 min in acid alcohol, washed with tap water, and then counter-stained for 5 min with methylene blue. Finally, the stained smear was washed with tap water, air dried, and examined under the light microscope (*Garcia et al., 2017*).

### Rapid chromatographic immunoassay test

A previously described, ImmunoCard STAT! CGE was performed according to manufacturer's instructions to detect *Giardia lamblia, Cryptosporidium*, and *Entamoeba histolytica* antigens by the monoclonal antibodies (*Alharbi et al., 2020*).

### Molecular diagnosis (real-time PCR)

The DNA was extracted from stool samples (stored at −20 °C) by using the QIAamp Fast DNA Stool Mini Kit as described by the manufacturer's instructions (*QIAamp, 2020*), then kept at −20 °C until use.

The extracted DNA was amplified and detected for STHs specifically. Target genes included the internal transcribed spacer (ITS1) was used for *A. lumbricoides* and *T. trichiura* assays, and the ITS2 for *A. duodenale* and *N. americanus* assays (GenBank accession nos. AJ000895, FM991956, AJ001594, and AJ001599, respectively) (*Taniuchi et al., 2011*; *Mejia et al., 2013*). Furthermore, the 18S ribosomal RNA (rRNA) gene was

used for *S. stercoralis* (GenBank accession no. AF279916) (*Taniuchi et al., 2011*). All the primer sequences (5′ to 3′) used in the present study are listed in (Table 1).

The master mix, working primers and probe solutions were prepared to the specifications previously described (*Alqarni, Wakid & Gattan, 2022*), and stored at −20 °C until use.

The reaction mix was dispensed in appropriate volume of 15 µl into each well of PCR reaction plate containing 96-well. Then 5 µl of the DNA was added to the individual wells containing the reaction mix. Negative control was used in in every run. The plate was then transferred into Applied Biosystems 7500 Fast real-time PCR system, to start the cycling program and perform data analysis. Real-time PCR conditions were set up according to QuantiTect® Probe PCR kit instructions as shown in (Table 2).

### Statistical analysis

Categorical data was reported as frequency, cross-tabulation, and percentage (%). Data was analyzed by using SPSS (version 25). Statistical significance for the variation in the frequency between groups were determined by Pearson chi-square $\chi2$ test. The *p*-value was calculated at a significance level of ($p < 0.05$).

## RESULTS

Stool samples were examined macroscopically, and the consistency of formed ($n = 73$) and soft to loose ($n = 115$) represented the samples with no abnormal color, or blood/mucus observed. In addition, there were no gross parasitic stages such as segments of tapeworms or adult worms in samples of any investigated worker. There was no statistically significant difference with the physical characteristics of the stool.

Out of 188 participants, 103 (54.8%) and 85(45%) were female and males respectively, with age range between 22–34 years (26.08 ± 2.67).

Among the analyzed 188 stool specimens by all methods (microscopy, rapid diagnostic test and real-time PCR), 35 samples (18.62%), were infected with intestinal parasites, of which some with multiple infections, including helminths (*T. trichiura*, hookworms (*N. americanus* or *A. duodenale*), *A. lumbricoides*, *S. stercoralis*, *Hymenolepis nana*, *Heterophyes heterophyes*), and protozoa (*E. coli*, *E. histolytica*, *B. hominis*, *G. lamblia*, *I. buetschlii*, *E. nana*), (see Table 3 and Fig. 1).

Microscopic examination of stained smears with Kinyoun's and rapid chromatographic immunoassay test revealed that none of the investigated newly arrived workers was infected with *Cryptosporidium*.

Our investigation of infection types in relation to socio-demographic status, nationality, education level, occupation, symptoms, and other factors are not included in the present study, and will be introduced as a separate study soon.

As shown in Table 4, real-time PCR was able to detect the DNA in (19/188, 10%) samples followed by Ritchie sedimentation technique (18/188, 9.6%) while direct smear was able to detect parasites in only (7/188, 3.7%) of the samples. No statistically significant difference was found among three techniques ($p > 0.05$).

**Table 1 Primers and probe for real-time PCR.** All the oligonucleotide sequences (5′ to 3′) used in the present study.

| Parasite name/primer | | Sequences |
|---|---|---|
| *A. lumbricoides* ITS-1 | F | GTAATAGCAGTCGGCGGTTTCTT |
| | R | GCCCAACATGCCACCTATTC |
| | P | TTGGCGGACAATTGCATGCGAT |
| *A. duodenale* ITS-2 | F | GAATGACAGCAAACTCGTTGTTG |
| | R | ATACTAGCCACTGCCGAAACGT |
| | P | ATCGTTTACCGACTTTAG |
| *N. americanus* ITS-2 | F | CTGTTTGTCGAACGGTACTTGC |
| | R | ATAACAGCGTGCACATGTTGC |
| | P | CTGTACTACGCATTGTATAC |
| *T. trichiura* ITS-1 | F | TCCGAACGGCGGATCA |
| | R | CTCGAGTGTCACGTCGTCCTT |
| | P | TTGGCTCGTAGGTCGTT |
| *S. stercoralis* 18S rRNA | F | GAATTCCAAGTAAACGTAAGTCATTAGC |
| | R | TGCCTCTGGATATTGCTCAGTTC |
| | P | ACACACCGGCCGTCGCTGC |

**Table 2 The real-time PCR conditions set up according to QuantiTect® Probe PCR kit.**

| Step | Initial activation | Denaturation | Combined annealing/extension |
|---|---|---|---|
| Temp | 95 °C | 94 °C | 60 °C |
| Time | 15 min | 15 s | 1 min |
| Cycling | 1 cycle | 45 cycles | |

## DISCUSSION

Many studies on the prevalence of intestinal parasite infections have been done, while a limited number focused on foreign workers in the Kingdom of Saudi Arabia. To the best of our knowledge, this study is the first in Jeddah to investigate the intestinal parasites among newly arrived expatriate labors in Jeddah.

Details of the infection in relation to workers socio-demographic status, nationality, level of education, occupation, symptoms, and other factors are not included here and will be presented soon in a separate study.

In the present study, microscopic examination of direct smear preparations was carried out, followed by the formal ether sedimentation method. This concentration technique enhances the chance of recovery for the diagnostic stages of parasites, mainly in light infections. In addition, real-time PCR was used as a highly sensitive and rapid utility in the detection and identification of STHs (*O'Connell & Nutman, 2016*; *Cunningham et al., 2018*).

**Table 3 Number of cases infected with single, double, and triple intestinal parasites including helminths and protozoa.**

| Type of infection | Detected parasites | No. of cases |
|---|---|---|
| Single (26) | *E. coli*[1] | 7 |
| | *A. lumbricoides*[1,2] | 4 |
| | *T. trichiura*[1,2] | 3 |
| | *N. americanus*[1,2] | 2 |
| | *S. stercoralis*[1,2] | 2 |
| | *G. lamblia*[1] | 2 |
| | *A. duodenale*[1,2] | 1 |
| | *E. histolytica*[1] | 1 |
| | *H. heterophyes*[1] | 1 |
| | *H. nana*[1] | 1 |
| | *E. nana*[1] | 1 |
| | *B. hominis*[1] | 1 |
| Double (8) | *T. trichiura*[1,2] + *N. americanus*[1,2] | 2 |
| | *N. americanus*[1,2] + *B. hominis*[1] | 1 |
| | *N. americanus*[1,2] + *E. histolytica*[1] | 1 |
| | *E. coli*[1] + *E. histolytica*[1] | 1 |
| | *E. histolytica*[1] + *B. hominis*[1] | 1 |
| | *S. stercoralis*[1,2] + *A. duodenale*[1,2] | 1 |
| | *T. trichiura*[1,2] + *S. stercoralis*[2] | 1 |
| Triple (1) | *T. trichiura*[1,2] + *E. coli*[1] + *I. buetschlii*[1] | 1 |

Notes:
[1] using microscopy.
[2] using real-time PCR.

**Helminths**

**Protozoa**

**Figure 1 Detected microscopic stages of intestinal parasites.** Detected microscopic stages of intestinal parasites including, (A) *T. trichiura* egg; (B) hookworm egg; (C) *A. lumbricoides* egg, (D) *S. stercoralis* rhabditiform larva; (E) *H. nana* egg; (F) *H. heterophyes* egg; (G) *E. coli* cyst; (H) *E. histolytica* cyst; (I) *B. hominis*, (J) *G. lamblia* cyst; (K) *I. buetschlii* cyst; (L) *E. nana* cyst.

**Table 4 Summary for the detection of STHs using three techniques (PCR, Ritchie technique, and direct smears).**

|  | Real-time PCR | Ritchie | Direct smears | *p*-value |
|---|---|---|---|---|
| *Positive* | 19 (10.1%) | 18 (9.6%) | 7 (3.7%) | >0.05 |
| *Negative* | 169 (89.9%) | 170 (90.4%) | 181 (96.3%) | |

Other supportive methods including permanent stains and a rapid immunochromatographic kit were used. Permanent trichrome is the most common stain used in parasitology for illustration of morphological features of cysts and trophozoites of intestinal amoebas, flagellates and ciliates. As trophozoites quickly perish, and immediate processing of samples is not always feasible, then permanent staining is the choice.

The prevalence of intestinal parasites in this study was 18.6% among the investigated 188 newly arrived workers in Jeddah. Previous studies conducted among foreign workers in Saudi Arabia reported various prevalence values. Studies in Bahrah, Makkah, Al-Madina, Al-Baha and Al-Khobar reported a prevalence of 22.3%, 16%, 15%, 54%, and 31.4%, respectively (*Wakid, 2020a*; *Ahmed et al., 2015*; *Taha, Soliman & Banjar, 2013*; *Mohammad & Koshak, 2011*; *Abahussain, 2005*).

Our study revealed that the total infected cases with intestinal worms were 25 (23 cases with nematoda, one with trematoda and one with cestoda). Some of these cases revealed multiple infections, while the most abundant STHs parasites among the workers were *T. trichiura*, *N. americanus*, *S. stercoralis*, and *A. lumbricoides*. Higher rates of STHs infection were detected in soft and loose samples. This is in agreement with the previous two studies in Al-Madinah that showed *T. trichiura*, *N. americanus*, and *A. lumbricoides* accounted for the highest rates of infections (*Taha, Soliman & Banjar, 2013*; *Imam, Abdulbaqi & Fahad, 2015*). According to Bahrah study, all types of STHs were isolated from the investigated samples (*Wakid, 2020a*).

Real-time PCR in this study showed higher ability to detect the STHs in comparison with microscopic techniques but with no significant differences. This finding is consistent with a previous study conducted in the United Arab Emirates (*Al-Rifai et al., 2020*), which is due to the differences in sensitivity, specificity, and speed of detection (*Ricciardi & Ndao, 2015*).

*H. heterophyes* was the only isolated trematoda in our study. On the other hand, *H. nana* was the only detected tapeworm, which was detected in the previous study in Bahrah (*Wakid, 2020a*).

Among the isolated protozoan parasites from twenty samples, we found that the pathogenic organisms (*E. histolytica* and *G. lamblia*) represented 30%, however the main detected parasites were *E. coli*, *E. histolytica*, and *B. hominis*. Furthermore, the studies in Al-Madinah reported that *G. lamblia*, *E. histolytica* and *E. coli* accounted for the highest protozoan infections (*Taha, Soliman & Banjar, 2013*; *Imam, Abdulbaqi & Fahad, 2015*). According to the study in Bahrah, the most prevalent isolated protozoa were *B. hominis, E. nana, E. coli, G. lamblia* and *E. histolytica* (*Wakid, 2020a*). *B. hominis* was found to be very

common in Makkah study representing 79% of the infected cases, and their finding in contrast to ours (*Ahmed et al., 2015*). In the past, *B. hominis* had been considered part of the intestinal flora, but recent clinical advances reveal that it is a controversial pathogen (*Badparva & Kheirandish, 2020*).

*Bahrami et al. (2019)* showed that direct microscopy is unable to distinguish *E. histolytica* from *E. dispar*. In our study we used the rapid diagnostic test that contains monoclonal antibodies specific for *E. histolytica*, which revealed that all detected four cases were *E. histolytica* and not *E. dispar* (*Van den Bossche et al., 2015*). Similarly, the detection of the two cases with *G. lamblia* was by microscopic methods and the rapid chromatographic immunoassay test.

Different species of non-pathogenic protozoa can infect humans, and don't cause symptoms. However, their presence in the stool is an important sign that the patient had ingested some fecal matter, and so their identification has a diagnostic value. In addition, these organisms should 'ring the alarm", because they have the same fecal-oral route of infection with the pathogenic organisms (*Wakid, 2020a*). In our study, three non-pathogenic parasites were identified, including *E. coli*, *I. buetschlii* and *E. nana*.

We believe that the role of large-scale screening among newly arrived expatriate workers by accurate clinical examination and suitable laboratory diagnostic techniques have a significant impact on the sensitivity of parasite identification and therefore on management and control programs of intestinal parasites including STHs.

## CONCLUSION

The infection of newly arrived foreign workers with intestinal parasites could give rise to a risk for the community. In the current study, almost 19% of the newly arrived expatriate labors had infection with 13 intestinal parasites. There is a need for continuous and regular surveys to handle the occurrence of these infections. For improvement of intestinal parasites identification, we recommend a supportive infrastructure for the application of concentration methods and molecular assays.

## ACKNOWLEDGEMENTS

The authors are very thankful to King Fahd Medical Research Centre and the Special Infectious Agents Unit for providing the space for laboratory investigations on the collected samples.

### Funding

The authors received no funding for this work.

### Competing Interests

The authors declare that they have no competing interests.

## Author Contributions

- Mohammad F. Al-Refai conceived and designed the experiments, performed the experiments, analyzed the data, prepared figures and/or tables, and approved the final draft.
- Majed H. Wakid conceived and designed the experiments, performed the experiments, analyzed the data, prepared figures and/or tables, authored or reviewed drafts of the article, and approved the final draft.

## Human Ethics

The following information was supplied relating to ethical approvals (*i.e.*, approving body and any reference numbers):

The Ethics and Research Committee of the Faculty of Applied Medical Sciences, King Abdulaziz University, approved this study, and written informed consent was obtained from each participant.

## Data Availability

The raw data, findings related to another study including infection details in relation to workers sociodemographic status, nationality, education level, occupation, symptoms, and other factors, are available in Tables 1–5.

## Supplemental Information

Supplemental information for this article can be found online at http://dx.doi.org/10.7717/peerj.16820#supplemental-information.

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
