# Peer review of "Prevalence of intestinal parasites and comparison of detection techniques for soil-transmitted helminths among newly arrived expatriate labors in Jeddah, Saudi Arabia"

_PeerJ, doi:10.7717/peerj.16820_

## Round 0.1 · original submission · Major Revisions

I have completed my evaluation of your manuscript. The reviewers recommend reconsideration of your manuscript following major revision. I invite you to resubmit your manuscript after addressing the comments below. When revising your manuscript, please consider all issues mentioned in the reviewers' comments carefully: please outline every change made in response to their comments and provide suitable rebuttals for any comments not addressed. Please note that your revised submission may need to be re-reviewed.

Reviewer 1 ·

Basic reporting

The study describes several methods to identify and characterize intestinal parasites in stool samples of expatriate labors in Saudi Arabia. Tha authors were able to highlight the imporatance of the study to spread awareness of fecal contamination and reduce the chances of spreading soil-transmitted helminths (STH). But this manuscript needs significant imporvement in order to increase its appeal to broader audience and understandable for the readers.

Experimental design

Major comments:
1. Stained images of microscopic examination should be included describing the name of the idenfied parasite in stool samples. This will help the community to identify them in the context of endemic spread of these pathogens
2. Results should have sections and better explained
3. Names of statistical tests performed should be included e.g., students t-test for pairwise comparison or one-way ANOVA followed by Dunnett's test for multiple comparison. Or, whatever tests were performed.

Minor comments:
1. Line 161-197: should be shortened or removed if it is same as the manufacturer's protocol. This information should be available in the kit. The authors can just provide the kit details and mention any modification that was made.
2. Line 174: parenthesis should be removed "(ITS2)"
3. Line 178: say primes instead of "oligonucleotide sequences"
4. Line 234-236: justify your statement by including references about what study has been done to show types of infection based on demographic factors as mentioned

Validity of the findings

This study will spread awareness of testing for infection in across border incoming labors.

Additional comments

Overall more data display will support the claims made by the authors and will increase the depth of the manuscript.

Reviewer 2 ·

Basic reporting

In general, the manuscript is easy to understand and well-written.

Experimental design

1. The authors should include more details about the participant selection and sample collection process in this study in the section "Materials and Methods". For example, what is the study's duration? Are there any established criteria for participant selection, such as age, gender, and other relevant factors? Were all specimens examined simultaneously? If not, how the samples were stored and handled?

2. For experiments related to qPCR, what is the Ct value that is considered positive?

Validity of the findings

1. The authors claim that 25 cases were infected with intestinal helminth (line 223 and 269) with 2 positive samples for H. nana and H. heterophyes. However, in Table 5, the numbers of STH-positive cases diagnosed by PCR, Ritchie, and direct smears are 19, 18, and 7, respectively. Is it because some cases were only identified by a single diagnostic method? The lack of details makes these numbers confusing. The authors should provide additional information, such as the number and types of observations co-infections, whether there is any potential bias in the occurrence of co-infections and individual diagnostic outcomes for each of the 25 cases using each three diagnostic method. This also raises another question: What criteria did the authors employ to define an STH-positive case?

2. The authors should integrate the case numbers (Line 223-233) into Figure 1, rather than only presenting the percentage. Additionally, while the authors have indicated that the relationship between types of infections and factors will be presented in a separate study, they should at least mention or discuss how these infections are associated with the gender of the participants (related to line 217).

Additional comments

No comment

Reviewer 3 ·

Basic reporting

Clear and unambiguous, professional English used throughout. Literature references, sufficient field background/context provided.
The study titled Prevalence of intestinal parasites and comparison of detection techniques for soil-transmitted helminths among newly arrived expatriate labors in Jeddah, Saudi Arabia is a well-done investigation. The manuscript addresses public health issue of STHs and it significantly add-in the information about prevalence of intestinal parasites in expatriates in Saudi Arabia. The manuscript can be accepted in PeerJ however, the authors must address the following issues prior to the acceptance.
1. Please describe the need of project in the abstract which is deficient.
2. In Abstract, the conclusion needs to be improved in line of the results of this study.
3. These STHs belong to the Phylum Nematoda (roundworms), and include…..
4. Please modify the sentence as: These STHs belong to the Phylum Nematoda (roundworms), and a few examples include……
5. Line 70-72: Mention the reference
6. Line 80: Please delete South Asia as this falls under Asia itself.
7. If possible, please mention the name of laboratory of expatriate workers polyclinic
8. Authors should delete the extensive methodology of each step described under the heading Molecular Diagnosis (Real-time PCR).
9. There is no need for Table 2 and Table 3. Please delete. A little description in the text is sufficient.
10. Delete E. coli from the Figure 1 as this is not a STHs.
11. Is it possible for the authors to present the Figure 1 is some other attractive way? If yes, doing this will fetch attention of the readers.
12. Line 260-262 should be deleted.
13. Discussion section need to be improved with incorporation of multiple latest references.

Experimental design

Research question well defined, relevant & meaningful. It is stated how research fills an identified knowledge gap. Methods described with sufficient detail & information to replicate.
The experimental design is well explained. However, the following suggestions are needed to be incorporated/deleted from the manuscript.
1. If possible, please mention the name of laboratory of expatriate workers polyclinic
2. Authors should delete the extensive methodology of each step described under the heading Molecular Diagnosis (Real-time PCR).
9. There is no need for Table 2 and Table 3. Please delete. A little description in the text is sufficient.
10. Delete E. coli from the Figure 1 as this is not a STHs.
11. Is it possible for the authors to present the Figure 1 is some other attractive way? If yes, doing this will fetch attention of the readers.

Validity of the findings

Impact and novelty not assessed. Meaningful replication encouraged where rationale & benefit to literature is clearly stated. All underlying data have been provided; they are robust, statistically sound, & controlled. Conclusions are well stated, linked to original research question & limited to supporting results.

---

## Round 0.2 · Minor Revisions

I have completed my evaluation of your manuscript. The reviewers recommend reconsideration of your manuscript following minor revision. I invite you to resubmit your manuscript after addressing the comments below. When revising your manuscript, please consider all issues mentioned in the reviewers' comments carefully: please outline every change made in response to their comments and provide suitable rebuttals for any comments not addressed.

Reviewer 1 ·

Basic reporting

Please include scale bar in the microscopic images.

Experimental design

No comments.

Validity of the findings

No comments.

Reviewer 2 ·

Basic reporting

The authors have answered all my questions and I do not have any other comments.

Experimental design

No comments

Validity of the findings

No comments

Reviewer 3 ·

Basic reporting

Clear and unambiguous, professional English used throughout.

Experimental design

Research question well defined, relevant & meaningful.

Validity of the findings

Meaningful replication encouraged where rationale & benefit to literature is clearly stated.

---

## Round 0.3 · accepted · Accept

It is a pleasure to accept your manuscript entitled "Prevalence of intestinal parasites and comparison of detection techniques for soil-transmitted helminths among newly arrived expatriate labors in Jeddah, Saudi Arabia " in its current form for publication in PeerJ.